# Advances in Molecular Pathophysiology and Targeted Therapy for Cushing’s Disease

**DOI:** 10.3390/cancers15020496

**Published:** 2023-01-13

**Authors:** Shinobu Takayasu, Kazunori Kageyama, Makoto Daimon

**Affiliations:** Department of Endocrinology and Metabolism, Hirosaki University Graduate School of Medicine, 5 Zaifu-cho, Hirosaki 036-8562, Japan

**Keywords:** Cushing’s disease, adrenocorticotropic hormone, proopiomelanocortin, pituitary neuroendocrine tumor, corticotroph

## Abstract

**Simple Summary:**

Cushing’s disease is caused by autonomous adrenocorticotropic hormone (ACTH) produced by corticotroph pituitary neuroendocrine tumors, leading to cortisol over-production in the adrenal glands. Cushing’s disease presents with a variety of clinical features, ranging from overt to subtle, which are caused by hormonal activities. To treat the disease adequately, the molecular and genetic causes of Cushing’s disease need to be understood. This review discusses recent advances in molecular insights and targeted therapy for Cushing’s disease.

**Abstract:**

Cushing’s disease is caused by autonomous secretion of adrenocorticotropic hormone (ACTH) from corticotroph pituitary neuroendocrine tumors. As a result, excess cortisol production leads to the overt manifestation of the clinical features of Cushing’s syndrome. Severe complications have been reported in patients with Cushing’s disease, including hypertension, menstrual disorders, hyperglycemia, osteoporosis, atherosclerosis, infections, and mental disorders. Cushing’s disease presents with a variety of clinical features, ranging from overt to subtle. In this review, we explain recent advances in molecular insights and targeted therapy for Cushing’s disease. The pathophysiological characteristics of hormone production and pituitary tumor cells are also explained. Therapies to treat the tumor growth in the pituitary gland and the autonomous hypersecretion of ACTH are discussed. Drugs that target corticotroph pituitary neuroendocrine tumors have been effective, including cabergoline, a dopamine receptor type 2 agonist, and pasireotide, a multi-receptor-targeted somatostatin analog. Some of the drugs that target adrenal hormones have shown potential therapeutic benefits. Advances in potential novel therapies for Cushing’s disease are also introduced.

## 1. Introduction

Cushing’s disease presents with a variety of clinical features, ranging from overt to subtle, which are caused by hormonal activities. Autonomous adrenocorticotropic hormone (ACTH) produced by corticotroph pituitary neuroendocrine tumors induces cortisol over-production in the adrenal glands. As a result, severe complications have been observed in patients with Cushing’s disease, such as hypertension, menstrual disorders, hyperglycemia, osteoporosis, atherosclerosis, infections, and mental disorders. The primary treatment for Cushing’s disease is surgical excision of the tumor from the pituitary gland. However, curative surgery remains challenging due to the nature of the tumor, and additional therapies are required to treat tumor growth and the autonomous hypersecretion of ACTH. More potent medical treatments that target corticotroph pituitary neuroendocrine tumors have been developed. In order to treat Cushing’s disease adequately, it is necessary to understand its molecular and genetic causes. The pathophysiological characteristics of hormone production and tumor cell proliferation are also explained in this review.

In Cushing’s disease, the dysregulation of ACTH synthesis and secretion is caused by corticotroph tumors. Somatic mutations in the *ubiquitin-specific protease 8* (*USP8*) gene have been reported in corticotroph tumors [1,2,3] and play an important role in ACTH production. The mutation increases the enzyme’s activity, which results in excessive deubiquitination of epidermal growth factor receptor (EGFR) tyrosine kinase, disturbing its degradation [1]. EGFR expression levels are positively correlated with ACTH production and cell proliferation in corticotroph tumors. Overexpression of the cell-cycle regulator cyclin E and low expression levels of the cell-cycle inhibitor tumor protein 27Kip1 (p27) are found in Cushing’s disease [4,5]. Cyclin E expression is correlated with p27 loss in human corticotroph tumors [6]. Therefore, such cell-cycle regulators can be targeted for treatment.

Compared with normal corticotrophs, pituitary corticotroph tumors require higher doses of glucocorticoids to inhibit ACTH secretion. Therefore, corticotroph tumors are considered to have a partial lack of response to glucocorticoid negative feedback. Various factors have been suggested to be involved in this resistance. Corticotroph tumors overexpress heat shock protein 90 (HSP90), leading to disruption of the HSP90 complex, which induces the partial glucocorticoid resistance of corticotroph tumors. Testicular receptor 4 (TR4) interacts with glucocorticoid receptor (GR), leading to the disruption of GR binding to the *proopiomelanocortin* (*POMC*) promoter. BRG1, the ATPase component of the SWI/SNF chromatin remodeling complex, and histone deacetylase 2 (HDAC2) are involved in the resistance, because these factors are required for GR recruitment.

## 2. Molecular Pathophysiology

Cushing’s disease is caused by dysregulation of ACTH synthesis and secretion, due to corticotroph tumors. It is therefore important to understand the regulation of ACTH production and release, and the difference between normal corticotroph cells and corticotroph tumor cells. The murine AtT-20 corticotroph tumor cell line has been used as a model of normal corticotroph cells to evaluate molecular mechanisms of *Pomc*, the precursor of ACTH transcription and secretion, because no other corticotroph cell lines are available. Many aspects of the pathophysiology of Cushing’s disease have also been studied using AtT-20 cells, as human corticotroph tumors are rare and most tumors are relatively small. Differences can exist between species and between normal corticotroph cells and tumor cells, therefore, critical attention must be paid to the results. Three main factors are involved in the etiology of Cushing’s disease: tumorigenesis, autonomous ACTH production and secretion, and glucocorticoid resistance. These factors mutually affect each other. This section focuses on the molecular and genetic causes of Cushing’s disease.

### 2.1. Mutations in the Ubiquitin-Specific Protease 8 (USP8) Gene and Epidermal Growth Factor Receptor (EGFR) Expression

Somatic mutations in the *USP8* gene have been reported in 24–60% of corticotroph tumors [1,2,3]. These mutations are frequently observed in female patients and present as smaller tumors demonstrating strong *POMC* expression and ACTH production [1,2,3,7]. The mutations seem to play a significant role in ACTH production, rather than cell proliferation. However, a de novo germline mutation in the hotspot region of the *USP8* gene was discovered in pediatric patients with Cushing’s disease [8], with the report showing that the mutation was involved in corticotroph tumorigenesis. An in vitro study showed that this mutation increased enzyme activity, resulting in excessive deubiquitination of EGFR tyrosine kinase, disturbing its degradation and increasing its recycling [1]. Although the association between EGFR expression and *USP8* mutation status in Cushing’s disease is controversial [3,7], EGFR expression has been observed in human corticotroph tumors, and EGFR expression levels positively correlate with ACTH levels and tumor recurrence status [9,10]. Moreover, studies in vivo and in vitro have reported that EGFR overexpression was involved in corticotroph tumor formation, cell proliferation, *Pomc* expression, and ACTH secretion via mitogen-activated protein kinase (MAPK) activation, and these factors are inhibited by EGFR inhibitors [9,11,12]. Gefitinib, a tyrosine kinase inhibitor targeting EGFR, attenuated *POMC* expression and ACTH secretion in human and canine corticotroph tumor cell cultures [9].

Ex vivo or in vivo studies using USP8 inhibitors have been rarely reported, but some reports have shown that USP8 inhibitors decreased ACTH secretion and cell proliferation in AtT-20 cells [13,14]. For example, we reported that a potent USP8 inhibitor, DUBs-IN-2, decreased *Pomc* mRNA and ACTH levels in AtT-20 cells. Furthermore, DUBs-IN-2 also inhibited cell proliferation and induced apoptosis. Inhibition of the ubiquitin–proteasome pathway increased ACTH secretion and intracellular content in normal rat pituitary cells in primary cultures [15] as well as primary cultures of human corticotrope tumors, regardless of mutations [16]. These results suggest that *POMC* is a target of ubiquitylation, and the ubiquitin system is directly involved in ACTH synthesis and processing pathways in normal corticotroph cells and in tumor cells.

Deubiquitinase gene *USP48* mutations were identified in human corticotroph tumors by Chen et al. [17]. They reported that 23% (21/91) of the studied wild-type *USP8* cases harbored *USP48* (encoding p.M415I or p.M415V) mutations, while only 1.2% (1/78) of *USP8* mutated cases had *USP48* mutation. They also observed increased rodent *Pomc* promoter activity and *Pomc* mRNA expression in the mutated USP48 overexpressed-AtT-20 cells. Sbiera et al. [18] showed that *USP48* mutations are relatively frequent in *USP8* wild-type tumors, with in vitro experimentation demonstrating that this mutation increases catalytic activity, which leads to deubiquitination of tumor transcription factor Gli1. They also showed that overexpressed USP48 mutations enhanced *Pomc* promoter activity via Gli1 in AtT-20 cells. Further studies are needed to clarify how *USP48* mutation is involved in ACTH excess and tumorigenesis.

### 2.2. Cell Cycle Regulator

Overexpression of the cell-cycle regulator Cyclin E and low expression levels of the cell-cycle inhibitor p27 are observed in Cushing’s disease [4,5]. Additionally, expression levels of Cyclin E expression are related to p27 loss in human corticotroph tumors [6] (Figure 1). A combination of *cyclin E* overexpression with *p27* knockout increased the incidence, proliferation, and size of pituitary tumors in mouse models. Indeed, reports revealed that R-roscovitine, a potent inhibitor of cyclin-dependent kinase 2 (CDK2)/Cyclin E, inhibited ACTH expression in models of zebrafish (transgenic zebrafish with overexpression of the pituitary tumor transforming gene), mice (xenografted with AtT-20 cells), and primary cell cultures of surgically resected human corticotroph tumors [19,20].

The negative cell-cycle regulator *Cdk5 and ABL enzyme substrate 1* (*Cables1*) was found to be a glucocorticoid-dependent gene in AtT-20 cells, according to gene expression profiling [21]. CABLES1 inhibits Cyclin E, both directly and indirectly, resulting in suppression of the cell cycle (Figure 1). However, CABLES1 is completely absent in more than half of human corticotroph tumors [21]. Cyclin E overexpression, which is correlated with low expression levels of p27, may stimulate the cell cycle in tumor cells (Figure 1). Germline *CABLES1* missense mutations were identified in 2% of tested human corticotroph tumors, and none of the patients had somatic *USP8* mutations [22]. Mutations were observed in early-onset aggressive macrotumors [22]. In vitro experimentation showed that *CABLES1* mutations led to the loss of inhibitory effects on cell proliferation by glucocorticoids [22].

### 2.3. Genetic Familial Syndromes, Oncogenes, and Tumor Suppressor Genes

Genetic familial syndromes associated with pituitary tumors, such as familial isolated pituitary adenoma, multiple endocrine neoplasia, McCune–Albright syndrome, DICER1 syndrome, and tuberous sclerosis complex are rarely found in Cushing’s disease. Oncogenes and tumor suppressor genes that often occur in other tumor types are rarely found in corticotroph tumors. Chen et al. reported that 16.4% (15/91) of wild-type *USP8* cases included the *BRAF* p.V600E mutation [17]. They demonstrated that *Braf* mutations enhanced rodent *Pomc* promoter activity and *Pomc* mRNA expression in AtT-20 cells. Moreover, compared to wild-type cells, primary cultured human corticotroph adenoma cells with BRAF mutations displayed greater reduction of ACTH secretion in response to the BRAF inhibitor vemurafenib. It has been well established that *BRAF* p.V600E mutation drives the activation of the MAPK signaling pathway and that *BRAF* is frequently mutated in different cancer types, including leukemia, melanoma, thyroid carcinoma, colorectal adenocarcinoma, glioblastoma, and non-small-cell lung cancer [23,24]. It has also been well established that the hypothalamic corticotropin-releasing factor (CRF) stimulates the *Pomc* transcription and release of ACTH via the activated MAPK-nuclear receptor subfamily 4, group A (NR4A) pathway in AtT-20 cells [25]. Therefore, these results may suggest that *BRAF* mutation enhances *POMC* transcription and ACTH secretion via the MAPK pathway in Cushing’s disease. Some studies have reported that *BRAF* V600E variants are extremely rare in human corticotroph pituitary tumors [18,26,27]. However, mutations of the tumor protein 53 (*P53*) tumor suppressor gene seem to be more frequent than reported in previous data. Sbiera et al. showed that 33.3% (6/18) of studied cases had *P53* variants, especially in larger corticotroph tumors [18]. According to a meta-analysis, *P53* mutations were found in about 12% of ACTH-producing pituitary tumors [28].

### 2.4. Glucocorticoid Resistance

ACTH-producing pituitary tumors require higher doses of glucocorticoids to inhibit ACTH secretion compared with normal corticotrophs, while most ectopic ACTH-secreting tumors are not inhibited even by high doses of glucocorticoids [29,30]. Typical corticotroph tumors show a partial lack of response to glucocorticoid negative feedback, although the mechanism remains largely uncertain.

Somatic loss-of-function mutations in the *nuclear receptor subfamily 3*, *group C*, *member 1* (*NR3C1*) gene encodes the GR [31,32,33], and overexpression of 11β-hydroxysteroid dehydrogenase 2 is an enzyme that converts cortisol to inactive cortisone [34,35] have rarely been reported in corticotroph adenomas. GR interacts with NR4A families through protein–protein interactions, and GR represses various genes via a trans-repression mechanism [36]. BRG1, the ATPase component of the SWI/SNF chromatin remodeling complex, and histone deacetylase 2 are required to recruit GR [37] (Figure 2). BRG1 constitutively presents and behaves as a scaffold protein at the *Pomc* promoter. GR recruitment results in decreased histone acetylation, activating transcription at the *Pomc* promoter and gene body (Figure 2). Overall, the loss of BRG1 may cause glucocorticoid resistance. Furthermore, the loss of nuclear expression of either BRG1 or histone deacetylase 2 was reported in over 50% of corticotroph tumors in dogs or humans [37]. BRG1 may also work as a modulator of the cell cycle and tumorigenesis, as the loss of BRG1 in these corticotroph tumors is involved in the overexpression of cyclin E and loss of p27 [6].

The proposed model for the nuclear translocation pathways, including complex formation of GR, has been disputed. HSP90 was traditionally thought to act as one of the anchoring proteins that retain GR in the cytoplasm [38]. After glucocorticoid binding, the activated GR dissociates from the HSP90 complex and translocates to the nucleus [39]. However, it has also been reported that GR remains in the HSP90 complex and is transported into the nucleus after glucocorticoid binding-induced substitution of FK506-binding immunophilin 5 (FKBP5) for FKBP4, and recruitment of the transport protein dynein, resulting in GR converting to the DNA-binding form in the nucleus [40] (Figure 2). Riebold et al. [41] showed that HPS90 is overexpressed in corticotroph tumors compared with the normal pituitary gland and non-functioning pituitary tumors. The overexpressed HSP90 leads to the disruption of the complex, which induces the partial glucocorticoid resistance of corticotroph tumors. They also found that silibinin, a C-terminal HSP90 inhibitor, dissociates GR from HSP90 in AtT-20 cells. The inhibitor increases GR-dependent transcription activity or decreases *Pomc* transcription and ACTH secretion in AtT-20 cells. In addition, silibinin decreased ACTH levels in primary cultures of human corticotroph tumor cells and normal rat anterior pituitary cells. Furthermore, the inhibitor showed anti-tumorigenic effects in a nude mouse tumor model.

Although the expression levels of FKBP4 and FKBP5 in corticotroph tumors have not been determined, reduced leukocyte DNA methylation of *FKBP5* [42] and increased *FKBP5* mRNA levels in blood [43] have been found in patients with Cushing’s syndrome. In our previous study, we found that FKBP5 mRNA and protein levels were increased by glucocorticoids in AtT-20 cells [44]. Additionally, glucocorticoid-induced decreases in *Pomc* mRNA levels were further decreased by *Fkbp5* knockdown [44]. These results support the hypothesis that FKBP5 is involved in glucocorticoid resistance in corticotroph tumors.

Nuclear receptor subfamily 2, group C, member 2, also known as TR4, has been strongly expressed in human corticotroph tumors [45]. Overexpression of TR4 enhanced *Pomc* promoter activity, *Pomc* mRNA expression, and ACTH secretion in AtT-20 cells [45] and in primary cultures of human pituitary corticotroph tumors [46]. In addition, a TR4 overexpression tumor model in nude mice increased tumor growth rates and ACTH and corticosterone levels in the blood, whereas these were decreased in a TR4 knockdown model [45]. Co-immunoprecipitation studies performed in AtT-20 cells showed that TR4 interacts directly with the N-terminal domain of GR, leading to the disruption of GR binding to the *POMC* promoter (Figure 2). As a result, TR4 overexpression attenuated GR-mediated inhibition of *POMC* transcription in AtT-20 cells and human corticotroph tumor cells [46]. Therefore, overexpression of TR4 in human corticotroph tumors may contribute to glucocorticoid resistance in Cushing’s disease.

## 3. The Practice of Treatment

Surgical excision of the tumor from the pituitary is the primary treatment for Cushing’s disease. However, when curative surgery fails, additional therapies are required.

### 3.1. Surgery

The primary treatment for Cushing’s disease is surgical excision of the tumor from the pituitary gland. Microscopic or endoscopic transsphenoidal surgery (TSS) by a skilled neurosurgeon with rich experience is recommended. When no adenoma is found on the preoperative magnetic resonance imaging (MRI), technical specificity is especially required [47]. In about 65–90% of patients with a pituitary microtumor, remission is achieved as a result of the initial surgery [48]. When tumor growth and autonomous hypersecretion of ACTH are found after surgery, it is necessary for residual tumors to be treated by repeat pituitary surgery. If the second surgery is unsuccessful or inappropriate, alternative therapies such as medical and/or radiation therapy should be considered [49].

### 3.2. Radiotherapy

Radiotherapy helps reduce the hypersecretion of ACTH and cortisol, and is an effective second treatment option for Cushing’s disease when residual or recurrent tumors are growing. 

Stereotactic radiotherapy, such as Gamma Knife, Cyberknife, or proton beam radiotherapy should be used rather than conventional radiotherapy. Tumor growth is controlled in 95% of cases and hormones are controlled in 54–68% of cases where stereotactic radiotherapy is used, after a period of several months to a few years [50]. While awaiting the effects of the radiation, medical therapy to reduce and control cortisol levels is recommended.

Hypopituitarism, which may develop up to 10–15 years later, is the most common adverse effect of the treatment, and visual toxicity and cranial nerve neuropathy have also been reported. Therefore, long-term surveillance is mandatory after pituitary radiation [51].

### 3.3. Medical Therapy

For the treatment of Cushing’s disease, certain drugs such as cabergoline and somatostatin analogs that target corticotroph tumors in the pituitary have shown therapeutic benefits. Somatostatin receptor agonists have demonstrated inhibitory effects on hormone production and tumor cell proliferation in somatotroph and thyrotroph tumors. In Cushing’s disease, pasireotide has shown significant effects by decreasing the levels of ACTH and cortisol in the blood, and reducing levels of urinary free cortisol (UFC). Osilodrostat, which targets cortisol production from the adrenal glands, is a new treatment option for the disease.

#### 3.3.1. Clinical Management of Targeted Pituitary Therapy

##### Pasireotide

Pasireotide, a somatostatin analog, has a preferential affinity for somatostatin receptor type 5 (SSTR5), inhibiting ACTH secretion in patients with ACTH-secreting tumors [52]. Initially, pasireotide (600 or 900 µg) is injected subcutaneously twice daily. Monthly thereafter, long-acting pasireotide (10–40 mg) is injected intramuscularly. This treatment normalized UFC levels in about 40% of patients with Cushing’s disease [53]. The adenoma volume also decreased in more than half of patients [54]. SSTR-mediated inhibition of phosphorylated extracellular signal-related kinases (ERK) by pasireotide decreased the proliferation of corticotroph tumor cells [55]. Pasireotide may induce tumor shrinkage through direct action on tumor cells [56]. A recent report demonstrated that SSTR5 is more highly expressed in *USP8*-mutated tumors, and pasireotide exerts an increased antisecretory response in these tumors [57].

In patients with diabetes, blood glucose levels should be well controlled before the administration of pasireotide. Hyperglycemia-related adverse events occurred in 68.4–73.0% of patients with Cushing’s disease following subcutaneous pasireotide injection [52,58,59]. Hyperglycemia due to long-acting pasireotide (10 mg) was observed in 49% of patients [53]. This effect may be caused by reduced insulin and glucagon-like peptide-1 (GLP-1) secretion [60]. Pasireotide binds to SSTR5 in the pancreas, leading to reduced secretion of insulin and GLP-1-induced insulin [61,62]. However, a recent report of a randomized open-label Phase IV study showed that some patients treated with pasireotide developed hyperglycemia requiring antidiabetic treatment [63]. Nevertheless, for patients who develop hyperglycemia, metformin followed by incretin-based therapy could be effective [63]. Other adverse effects may include diarrhea or gallstones. Therefore, although the drug is effective, careful monitoring is required.

##### Cabergoline

Cabergoline is a dopamine receptor type 2 agonist. Initially, 0.5–1 mg is administered per os once a week at bedtime. The dose of cabergoline should be adjusted to maintain UFC excretion within the normal range [49]. About 40% of patients treated with cabergoline monotherapy achieved remission of Cushing’s disease [64]. Cabergoline at the maximal dose, ranging from 1–7 mg/week, higher than the dose usually administered in patients with hyperprolactinemia, achieved control of cortisol secretion [65]. Although treatment with cabergoline might be associated with an increased risk of cardiac valve disease, no correlation has been shown in patients with pituitary tumors [66,67].

Dopastatin, a chimeric somatostatin/dopamine compound, shows inhibitory effects on ACTH secretion in human corticotroph tumor cells [68]. However, highly potent dopaminergic metabolites were unfortunately produced after repeated administration in human subjects. As an alternative, BIM-23B065, a second-generation chimera, is currently being developed [69].

##### Temozolomide

Temozolomide is an orally active anti-neoplastic alkylating agent used in the treatment of highly aggressive brain tumors, principally glioblastomas. The drug is also effective for some aggressive atypical pituitary tumors and carcinomas, including Cushing’s disease [70]. However, in many countries’ insurance may not cover this drug [50]. Low expression levels of the DNA-repair enzyme O6-methylguanine-DNA methyltransferase (MGMT) may increase sensitivity to temozolomide, while high expression of MGMT protein predicts a poor response. However, preserving the function of DMA mismatch repair protein (MSH6) may contribute to the effectiveness of temozolomide [71]. In general, the traditional Stupp regimen has been adopted in most centers [72]: temozolomide 150–200 mg/m^2^ for a 5-day cycle every 28 days.

#### 3.3.2. Clinical Management of Targeted Adrenal Gland Therapy (Summary in Table 1)

##### Ketoconazole

Ketoconazole inhibits the cytochrome P450 family 17, subfamily A, member 1 (CYP17A1), CYP11A1, and CYP11B1 enzymes. Treatment is started at 200–600 mg/day and is typically continued at 400–1200 mg/day, administered in two or three divided doses (Table 1). Cytolytic hepatitis should be considered, and liver aminotransferases should be carefully monitored one week after treatment is initiated. Adverse events can occur, such as mild gastrointestinal upset or male hypogonadism due to a decrease in androgen synthesis. Therefore, the drug should preferentially be used in women, rather than in men [73]. Ketoconazole potently inhibits CYP3A4, which interacts with numerous drugs and affects drug metabolism. CYP3A4 inhibitors may increase ketoconazole levels. The efficacy of ketoconazole can be reduced when proton pump inhibitors are used or in patients with achlorhydria [74]. Ketoconazole can also prolong the QT interval, and electrocardiography (ECG) should be performed.

**Table 1 cancers-15-00496-t001:** Characteristics of drugs used for targeted adrenal gland therapy. CYP, cytochrome P450.

Drug Name	Interact with CYP Types	Dose Range	Adverse Effects	References
Ketoconazole	CYP17A1, 11A1, 11B1, 3A4	400–1200 mg	Cytolitic hepatitis, mild gastrointestinal upset, male hypogonadism, and QT prolongation	[73,74]
Metyrapone	CYP11B1, 11B2	500–3000 mg	Gastrointestinal upset, dizziness, nausea, hypokalemia, liver damage, and hirsutism	[75,76]
Osilodrostat	CYP11B1, 11B2	2–30 mg	QT prolomgation, fatigue, appetite loss, headache, nausea, dizziness, hypokalemia, hypertension, and hirsutism	[77,78]
Mitotane	CYP3A4	500–3000 mg	Gastrointestinal and neurological disorders, hypercholesterolemia, and hypothyroidism	[79]
Mifepristone	None	300–1200 mg	Hypokalemia, hypertension, edema, metrorragia, and endometrial hyperplasia	[80,81]

##### Metyrapone

Although metyrapone mainly inhibits CYP11B1, it also inhibits CYP11B2 enzymes. This drug has a rapid onset of action. Metyrapone therapy is used before surgery or as a second-line treatment after surgery or radiotherapy [75]. Initially, metyrapone (250 mg) is administered per os once daily, and the number of doses is subsequently increased every few days. Metyrapone should be administered in three to four daily divided doses. In a study by Daniel et al. [75], the median final dose of metyrapone was 1375 mg for the treatment of Cushing’s disease. In order to control cortisol levels satisfactorily, a few weeks of treatment may be required. Alternatively, enough metyrapone (500–3000 mg) is administered in three to four daily divided doses and then replaced with hydrocortisone (15–20 mg per day) (Table 1). This method may be recommended in patients whose severe hypercortisolemia is satisfactorily managed before surgery [49]. Metyrapone has been shown to reduce urinary secretion of cortisol and aldosterone, and more than 50% of patients achieved control of cortisol levels. Blood pressure, glucose levels, muscle strength, and mental manifestations also improved [74]. Adverse events with metyrapone, such as mild gastrointestinal upset or dizziness, occurred in 25% of patients [75]. Nausea, hypokalemia, and liver damage might be associated with these events. Adverse events due to hyperandrogenism may disallow long-term administration in female patients [76]. In Cushing’s disease, metyrapone treatment generally elevates ACTH levels in the blood; however, its effects on tumor size are unclear.

##### Osilodrostat

Osilodrostat is a new treatment option for Cushing’s disease. Osilodrostat inhibits CYP11B1 and CYP11B2 enzymes, resulting in cortisol inhibition. This drug is a more potent inhibitor and has a prolonged half-life compared with metyrapone. Similar to metyrapone, osilodrostat is used before surgery and is approved as a second-line treatment for patients not cured by pituitary surgery or those for whom pituitary surgery is inappropriate [74]. Osilodrostat is initiated at 2 mg per os twice per day and increased by 1–2 mg every 2 weeks until normalization of UFC levels or a maximal dose of 30 mg twice daily (Table 1). Alternatively, after sufficient amounts of osilodrostat have been administered, it is subsequently replaced with hydrocortisone (15–20 mg per day). Twice daily osilodrostat rapidly reduces UFC and is generally well tolerated [77]. Osilodrostat rapidly normalized UFC excretion in most patients with Cushing’s disease in Phase III trials [77,78]. Liver aminotransferase levels may be mildly increased, and associated adverse events include long QT syndrome, fatigue, appetite loss, headache, nausea, dizziness, hypokalemia, hypertension, and hirsutism.

##### Mitotane

Mitotane is commonly used in patients with adrenocortical cancer, yet is rarely used for Cushing’s disease. This drug causes irreversible change in adrenocortical cells. Mitotane should not be used before pituitary surgery. Mitotane is administered per os in three daily divided doses (250–500 mg), and doses are subsequently increased (Table 1). The drug suppresses hypercortisolism, but acts very slowly. When cell damage is induced by mitotane, replacement with hydrocortisone (15–20 mg per day) is required.

Mitotane also potently affects CYP3A4, which interacts with numerous drugs and affects drug metabolism. When adrenal insufficiency is treated or a block-and-replacement therapy is selected, glucocorticoid replacement doses two to three times higher than the normal dose are required [79].

##### Mifepristone

Mifepristone is a competitive glucocorticoid receptor antagonist that inhibits cortisol binding to the receptor. Mifepristone (300–1200 mg) is administered per os twice daily (Table 1). Administration of mifepristone increases ACTH and cortisol levels. Therefore, it is difficult to assess its efficacy and the appropriate drug titration. Clinical indicators such as blood glucose and blood pressure should be carefully monitored. Hypokalemia, hypertension, and edema are associated adverse events. Metrorrhagia and endometrial hyperplasia may occur in female patients, secondary to progesterone receptor antagonism. Treatment with mifepristone alters thyroid hormone levels, and levothyroxine replacement should be initiated; close monitoring and adjustment of thyroid hormone replacement are required [80]. Long-term treatment with mifepristone increases ACTH levels in approximately two-thirds of patients with Cushing’s disease. Tumor regrowth and volume progression of macroadenomas have also been reported [81].

#### 3.3.3. Advances in Novel Potential Therapy (Summary in Table 2)

Medical therapy for Cushing’s disease remains challenging because of a lack of definitive targeted pituitary therapy. More effective medical treatment is required that targets pituitary ACTH-producing tumors. Some new drugs including USP8, EGFR, HSP90, and HDAC inhibitors have shown potential therapeutic benefits. Further in vivo and human corticotroph tumor cell studies are required to determine these benefits.

##### USP8 Inhibitors

Somatic mutations in *USP8* have been identified in corticotroph tumors of Cushing’s disease [1,3,7]. As previously mentioned, the mutations hyperactivate USP8 and prevent EGFR degradation, leading to increased EGFR stability and expression, resulting in enhanced EGFR signaling in Cushing’s disease. *USP8* mutations also increase the promoter activity of POMC by stabilizing EGFR signaling. Bortezomib, a proteasome inhibitor, has been used to treat multiple myeloma [82], and the inhibition of USP8 by the drug has been effective in overcoming gefitinib resistance in non-small cell lung cancers [83]. The compound 9-oxo-9H-indeno[1,2-b]pyrazine-2,3-dicarbonitrile (DUBs-IN-2) and RA-9 are potent deubiquitinase enzyme inhibitor of USP8 [84]. The USP8 inhibitors decrease *Pomc* mRNA levels and ACTH levels in AtT-20 cells, and additionally inhibits cell proliferation and induces apoptosis in AtT-20 cells [13,14] (Table 2). Therefore, USP8-targeted therapy is a promising treatment for Cushing’s disease. USP8 inhibitors had potent effects on ACTH production and cell proliferation in mouse corticotroph tumor AtT-20 cells, and it should be explored whether this drug shows similar effects in human corticotroph tumors.

**Table 2 cancers-15-00496-t002:** Characteristics of drugs introduced for novel potential therapy. USP8, ubiquitin-specific protease 8; EGFR, epidermal growth factor receptor; HSP90, heat shock protein 90; ACTH, adrenocorticotropic hormone.

Specific Inhibitors	Drug Name	Dose Range	Possible Adverse Effects	References
USP8	DUBs-IN-2	Unknown	Not reported	[13,14]
Tyrosine kinase or EGFR	Gefitinib, lapatinib, and SD-1029	Unknown	Interstitial pneumonia, severe diarrhea, and hepatitis	[9,12,85,86]
HSP90	Silibinin and CCT018159	Unknown	Severe diarrhea, and eye disorders	[41,87]
Histone deacetylase	Trichostatin A, suberoylanilide hydroxamic acid, romidepsin, and tubastatin A	Unknown	Dehydration, hyperglycemia, and thrombocytopenia	[88,89,90,91,92]
Cyclin-dependent kinase	R-roscovitine	Unknown	Pancytopenia, nausea, and diarrhea	[19,20]
ACTH neutralizing antibody	ALD1613	Unknown	Not reported	[93]
Immune checkpoint inhibitors	Ipilimumab and nivolumab	1 and 3 mg/kg BW	Immune-related adverse events	[94]

##### Tyrosine Kinase or EGFR Inhibitors

Phosphorylated EGFR expression is reported to be found in most corticotroph tumors [95], and an EGFR inhibitor could be an effective treatment for EGFR-related corticotroph tumors. Gefitinib, one of the better known EGFR tyrosine kinase inhibitors, was shown to suppress ACTH production and tumor growth in an experimental mouse model of Cushing’s disease with AtT-20 allografts [9] (Table 2). As mentioned above, mutations of USP8 prevent EGFR degradation [1]. Thus, increased EGFR stability and expression levels contribute to the activation of EGFR signaling in Cushing’s disease (Figure 3).

Lapatinib, an active specific tyrosine kinase receptor inhibitor of both EGFR and p185her2/neu (HER2), shows broad-spectrum anti-tumor activity [85]. Lapatinib decreases *Pomc* mRNA and ACTH secretion, inhibits cell proliferation, and induces apoptosis in AtT-20 cells (Table 2). Lapatinib also decreased tumor weight in mice with AtT-20 allografts in vivo. Lapatinib decreases *Pomc* and *Pttg1* mRNA levels in the tumor and ACTH and corticosterone levels in the blood. Thus, lapatinib has significant effects on decreases in ACTH production and the proliferation of corticotroph tumor cells. Such EGFR-targeted therapies might be a relevant treatment for Cushing’s disease.

The Janus kinase/signal transducer and activator of transcription (Jak/STAT) is located downstream of EGFR signaling [96] and is known to be essential for EGFR-driven migration and invasion [97] (Figure 3). Among STATs, STAT3 is required for *Pomc* transcription [98], and it directly regulates the *Pomc* gene in the pituitary and hypothalamus [98,99]. The tolerance of normal cells to the loss of Jak2/STAT3 function has driven recent research efforts to identify molecules capable of Jak2/STAT3 inhibition [100], yielding SD-1029 as a novel potent Jak2 and STAT3 inhibitor. EGFR-dependent Jak2/STAT3 therapy has potential clinical implications for treating Cushing’s disease. It has been reported that SD-1029 decreases STAT3 phosphorylation in corticotroph AtT-20 tumor cells. The drug also decreases ACTH synthesis and cell proliferation and induces apoptosis in AtT-20 cells [86] (Table 2). Accordingly, a therapy that targets the EGFR-dependent Jak2/STAT3 pathway would have potential applications for treating Cushing’s disease.

##### HSP90 Inhibitors

HSP90 is an essential molecular chaperone related to the folding and stabilization of client proteins, which regulate the survival of cancer cells. HSP90 protein is overexpressed in Cushing’s disease; therefore, an appropriate HSP90 inhibitor would be useful for the effective treatment of Cushing’s disease [41]. As mentioned above, overexpression of HSP90 restrains the release of mature GRs, contributing to partial glucocorticoid resistance [41]. A C-terminal HSP90 inhibitor, silibinin, has been reported to restore glucocorticoid sensitivity [41] (Table 2). In our previous study, the HSP90 inhibitors 17-allylamino-17-demethoxygeldanamycin and CCT018159 suppressed ACTH production in corticotroph tumor AtT-20 cells [87] (Table 2). Treatment with CCT018159, a cell-permeable pyrazole resorcinol compound that inhibits the ATPase activity of HSP90, decreased tumor *Pomc* mRNA levels in mice with AtT-20 allografts. Plasma corticosterone levels also decreased following treatment with CCT018159 in vivo [87].

##### Histone Deacetylase Inhibitors

Trichostatin A inhibits ACTH synthesis by modulating its precursor *Pomc*, and blocks cell proliferation [88] (Table 2). Suberoylanilide hydroxamic acid, a pan-histone deacetylase (HDAC) inhibitor, also inhibited ACTH production and cell proliferation in AtT-20 and human corticotroph tumors [89] (Table 2). Romidepsin, a selective inhibitor of HDAC1/2, is known to induce cell-cycle arrest, cell apoptosis, and altered gene expression in malignancies [90]. Romidepsin decreased ACTH synthesis and inhibited cell proliferation via PTTG1 in murine corticotroph tumor AtT-20 cells [91]. HDAC1 may be involved in the proliferation of corticotroph cells via PTTG1, because *Hdac1* knockdown decreases basal *Pttg1* mRNA and cell proliferation. Romidepsin may therefore be a potential candidate as a therapy for Cushing’s disease. HDAC6 regulates HSP90 acetylation and modulates HSP90/glucocorticoid receptor protein–protein interactions. Tubastatin A, a selective HDAC6 inhibitor, also demonstrated inhibitory effects on *Pomc* and *Pttg1* mRNA expression [92]. Phosphorylated Akt levels are increased by tubastatin A. The Akt-PTTG1 pathway is involved in the regulation of cell proliferation.

Inhibition of the phosphoinositide-3-kinase (PI3K) pathway prevents cell viability in corticotroph tumors. Combined treatment with a PI3K inhibitor and an HDAC inhibitor additionally decreases ACTH production and cell proliferation. A dual HDAC and PI3K inhibitor, CUDC-907, has been developed, and therefore also offers a potential therapeutic option for Cushing’s disease [101].

##### Cyclin-Dependent Kinase Inhibitors

The cAMP-protein kinase A-cyclic AMP-response element-binding protein (CREB) pathway plays an important role in corticotroph cells. The protein kinase C (PKC) pathway increases intracellular Ca^2+^ and contributes to ACTH secretion in corticotroph tumor cells [102]. POMC gene expression is regulated via the PKC pathway in AtT-20 cells, but not in anterior pituitary cells [103]. Therefore, PKA and PKC pathways might both participate in gene regulation in corticotroph tumor cells via CREB phosphorylation.

Tumor proteins p27 and p53 serve as tumor suppressors. As a critical cell-cycle regulator, p27 arrests cell division and inhibits the G1/S transition. Cyclin E forms a complex with CDK2, and cyclin E/CDK2 complexes are involved in multiple cellular processes (Figure 3). The cyclin E/CDK2 complexes play a critical role in the G1 phase and in the G1/S phase transition, and the cell cycle is negatively regulated by p27 [104]. In our previous study, the DNA polymerase inhibitor aphidicolin increased p27 and p53 levels, while reducing levels of cyclin E. Both p27 and p53 regulate cyclin-dependent proteins, resulting in suppression of the cell cycle [105]. Fluorescence-activated cell sorting (FACS) analyses revealed that this drug decreased the percentage of S-phase cells and increased the percentage of G0/G1-phase cells in AtT-20 cells [105]. Aphidicolin can therefore inhibit cell proliferation via G0/G1 cell arrest in AtT-20 cells.

R-roscovitine (seliciclib or CYC202), a pharmacologic CDK2/cyclin E inhibitor, decreased ACTH and corticosterone levels in the blood when taken orally and restrained tumor growth in a mouse model of corticotroph tumors [19] (Table 2). R-roscovitine disrupts cyclin E and E2F transcription factor 1 binding to the *Pomc* gene promoter. R-roscovitine inhibits *POMC* expression in human corticotroph tumor cells [20]. 

##### ACTH-Neutralizing Antibody

A specific and high-affinity ACTH-neutralizing monoclonal antibody (ALD1613) has been generated [93]. ALD1613 inhibited ACTH-induced accumulation of cyclic adenosine monophosphate in rodent adrenal cells [93] (Table 2). In rodent models with acute restraint-stress-induced increases in ACTH levels, ALD1613 reduced corticosterone levels in the blood [93]. Therefore, ALD1613 may be a potentially effective therapy to suppress cortisol production in Cushing’s disease.

##### Immunotherapy

In corticotroph carcinomas and aggressive corticotroph tumors, the use of immune checkpoint inhibitors (ICIs) such as ipilimumab and nivolumab has been reported (Table 2). Plasma ACTH levels, pituitary mass, or liver metastasis may be decreased. Therefore, ICIs may be a promising therapeutic option for aggressive pituitary tumors and carcinomas [94].

## 4. Discussion and Conclusions

Drugs that target pituitary ACTH-secreting tumors, such as pasireotide, a new somatostatin analog, have been effective in the treatment of Cushing’s disease. However, hyperglycemia-related adverse events occur in most patients with Cushing’s disease who are treated with pasireotide. Therefore, blood glucose levels should be well controlled before administration of this drug. Osilodrostat is a new treatment option for the disease, targeting cortisol production from the adrenal glands. With appropriate attention to certain side effects described above, osilodrostat can be used before surgery and is approved as a second-line treatment for patients not cured by pituitary surgery.

More effective medical treatments that target the molecular and genetic causes of corticotroph pituitary neuroendocrine tumors are expected. The *USP8* gene mutation is involved in corticotroph tumorigenesis. This mutation increases enzyme activity and results in excessive deubiquitination of EGFR tyrosine kinase, disturbing its degradation and increasing its recycling. Therefore, USP8-targeted therapy offers a promising approach for the treatment of Cushing’s disease. In addition, EGFR-targeted therapy could have clinical applications for treating Cushing’s disease.

In Cushing’s disease, overexpression of the HSP90 protein has been observed, and this overexpression restrains the release of mature glucocorticoid receptors, leading to partial glucocorticoid resistance. Silibinin, a C-terminal HSP90 inhibitor, restores glucocorticoid sensitivity, therefore, an appropriate HSP90 inhibitor would be an effective treatment. Some histone deacetylase inhibitors have been shown to block cell proliferation and ACTH synthesis in AtT-20 and human corticotroph tumors. In vivo studies are required to determine these therapeutic benefits. R-roscovitine, a pharmacologic CDK2/cyclin E inhibitor, was also shown to restrain tumor growth in a mouse model of corticotroph tumors. The possible mechanisms are described above. Therefore, developments involving HSP90, histone deacetylase, and cyclin-dependent kinase inhibitors are expected. A specific and high-affinity neutralizing monoclonal antibody to ACTH (ALD1613) and the use of ICIs are also potential candidates for the treatment of Cushing’s disease.

In conclusion, drugs that target corticotroph pituitary neuroendocrine tumors, such as multi-receptor targeted somatostatin analogs, have been proven to be effective. New drugs that target adrenal hormones have shown potential therapeutic benefits. The development of novel potential clinical therapies for Cushing’s disease is also expected.

## Figures and Tables

**Figure 1 cancers-15-00496-f001:**
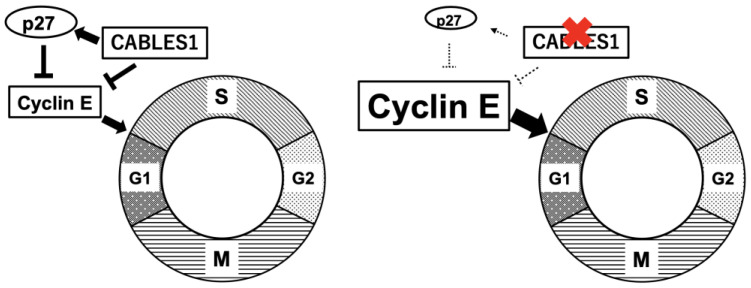
Cell-cycle regulators in normal corticotroph cells (**left panel**) and corticotroph tumor cells (**right panel**). The negative cell-cycle regulator Cdk5 and ABL enzyme substrate 1 (CABLES1) inhibit cyclin E directly and indirectly, resulting in the suppression of the cell cycle. However, CABLES1 is completely absent in more than half of human corticotroph tumors. *Cyclin E* overexpression, which correlates with low expression levels of the cell-cycle inhibitor tumor protein 27Kip1 (p27) may stimulate the cell cycle in tumor cells.

**Figure 2 cancers-15-00496-f002:**
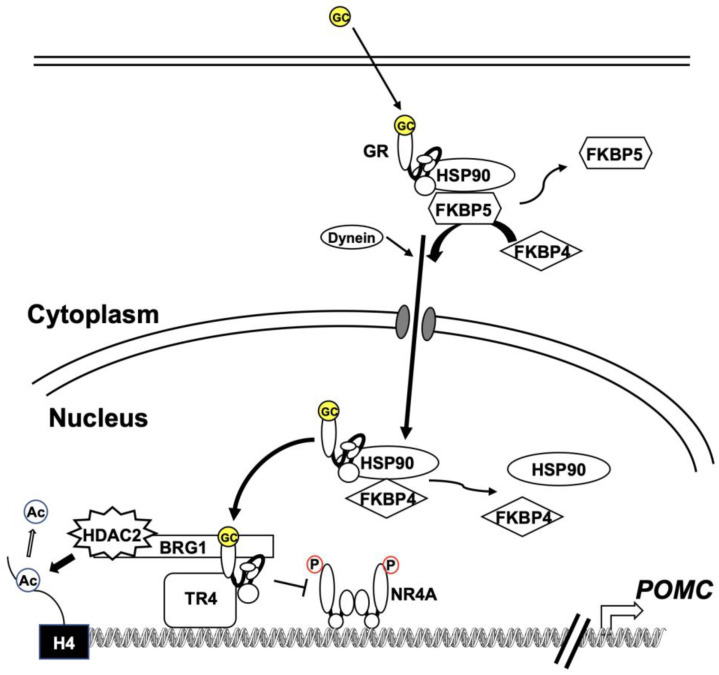
Putative mechanism of glucocorticoid resistance in Cushing’s disease. Corticotroph tumors overexpress heat shock protein 90 (HSP90), leading to disruption of the complex. Strong expression of testicular receptor 4 (TR4) interacts directly with glucocorticoid receptor (GR), which leads to the disruption of GR binding to the proopiomelanocortin (POMC) promoter. GR recruitment requires BRG1, the ATPase component of the SWI/SNF chromatin remodeling complex. Histone deacetylase 2 (HDAC2) results in decreased histone acetylation, which activates POMC transcription. The loss of nuclear expression of either of these proteins, BGR1 or HDAC2, is observed in more than 50% of human corticotroph tumors.

**Figure 3 cancers-15-00496-f003:**
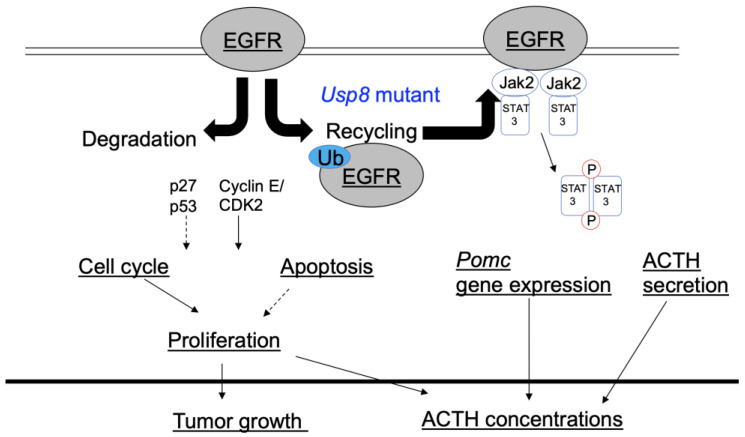
Proposed signaling mechanisms of the EGFR-related pathway in corticotroph tumor cells. The mutations hyperactivate ubiquitin-specific protease 8 (USP8), resulting in the inhibition of epidermal growth factor receptor (EGFR) degradation. Thus, increased EGFR stability contributes to the maintenance of EGFR signaling in Cushing’s disease. The Janus kinase/signal transducer and activator of transcription (Jak/STAT) is located downstream of EGFR signaling and is essential for EGFR-driven migration and invasion. Phosphorylated STAT3 is required for *Pomc* transcription. Ub; ubiquitin.

## Data Availability

Data are contained within the article.

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
