# Peer review of "Advances in Molecular Pathophysiology and Targeted Therapy for Cushing’s Disease"

_cancers, 2023, doi:10.3390/cancers15020496_

Round 1

Reviewer 1 Report

This is a narrative review that discusses in great detail the most recent advances in molecular insights of corticotroph tumors aiming for a brief perspective into future possible new target therapies for Cushing’s disease. The manuscript is well designed and nicely written and provides a complete overview of the current knowledge and potential candidates for the treatment of Cushing’s disease. I find no significant flaws to this manuscript and in my opinion it can be considered for publication.

Author Response

We appreciate your review. 

Reviewer 2 Report

Shinobu et al., describes ‘Advances in Molecular Pathophysiology and Target Therapy of Cushing’s Disease’. The authors then discuss about the recent advances in molecular insights and target therapy of Cushing’s disease. The review presented in this manuscript is well rationalized and interpreted.

Following are the suggestions to the authors.

1.      The introduction is too short, please elaborate further what are the pharmacological mechanism behind this type of disease.

2.      On page no.2, line 82 authors stated ‘Ex vivo or in vivo studies using USP8 inhibitors are rarely reported, but some reports showed that USP8 inhibitors decreased ACTH secretion and cell proliferation in AtT-20 cells’. Please explain in detail about these findings.

3.      Section 3.3.2, Clinical Management of Targeted Adrenal Gland Therapy; make a table (format below) including all the drugs which are used for treatment.

Drug name

Which CYP

Dose range

What are the adverse effect

 References

Similarly, for next section as well.

Specific inhibitor if any

Drug name

Which CYP

Dose range

What are the adverse effect

 References

4.      Discussion is missing in the manuscript, write few important points about these findings so that readers will be benefitted from this review.

Author Response

Following are the suggestions to the authors.

  1. The introduction is too short, please elaborate further what are the pharmacological mechanism behind this type of disease.

>The introduction was revised to further elaborate the pharmacological mechanism behind this type of disease.

  1. On page no.2, line 82 authors stated ‘Ex vivo or in vivo studies using USP8 inhibitors are rarely reported, but some reports showed that USP8 inhibitors decreased ACTH secretion and cell proliferation in AtT-20 cells’. Please explain in detail about these findings.

>The findings were further explained in detail (Page 3, line 132).

  1. Section 3.3.2, Clinical Management of Targeted Adrenal Gland Therapy; make a table (format below) including all the drugs which are used for treatment.

Drug name

Which CYP

Dose range

What are the adverse effect

 References

Similarly, for next section as well.

Specific inhibitor if any

Drug name

Which CYP

Dose range

What are the adverse effect

 References

>Following the Reviewer’s suggestion, two tables (Table 1 and Table 2) were added in the paper.

  1. Discussion is missing in the manuscript, write few important points about these findings so that readers will be benefitted from this review.

> Thank you for your suggestion. Important points about the findings in the paper have been mentioned in the discussion.
